# An Invex Relaxation Approach for Minimizing Polarization from Fully and Partially Observed Initial Opinions

## Abstract

This paper investigates the problem of minimizing polarization within a network, operating under the foundational assumption that the evolution of underlying opinions adheres to the most prevalent model, the Friedkin-Johnson (FJ) model. We show that this optimization problem under integrality constraints is $\mathcal{NP}$-Hard. Furthermore, we establish that the objective function fits into a specialized category of nonconvex functions called invex, where every local minimum is a global minimum. We extend this characterization to encompass a comprehensive class of matrix functions, including those pertinent to polarization and multiperiod polarization, even when addressing scenarios involving stubborn actors. We propose a novel nonconvex framework for this class of matrix functions with theoretical guarantees and demonstrate its practical efficacy for minimizing polarization without getting stuck at local minima. Through empirical assessments conducted in real-world network scenarios, our proposed approach consistently outperforms existing state-of-the-art methodologies. Moreover, we extend our work to encompass a novel problem setting that has not been previously studied, wherein the observer possesses access solely to a subset of initial opinions. Within this agnostic framework, we introduce a nonconvex relaxation methodology, which provides similar theoretical guarantees as outlined earlier and effectively mitigates polarization.

## 1 Introduction

In recent times, there has been a notable surge in the utilization of social media, accompanied by its increasingly pivotal role in shaping the discourse of global politics. Prominent social networks such as Twitter, Mastodon, Reddit, and others have emerged as influential platforms for users to articulate their viewpoints and participate in socio-political dialogues. Ironically, the original intention of social media to foster connectivity among individuals has, at times, yielded an unintended consequence: the emergence of echo chambers. This phenomenon arises from the preferential attachment behavior exhibited by users who tend to associate with others of similar inclinations, including shared political beliefs, as elucidated by Adamic & Glance (2005). Consequently, this trend has culminated in the polarization of active users within social media platforms along partisan lines, which, in turn, poses a potential threat to democratic ideals. The exposure of individuals primarily to like-minded peers serves to reinforce their preexisting convictions, a phenomenon identified by Cass (2002). This reinforcement of congruent perspectives, in turn, steers users toward confirmation bias, inadvertently increasing the polarization of the network (Kahneman, 2011).

Polarization within the realm of social networking platforms can be attributed to a complex interplay between an individual's actions and the underlying social algorithms governing the provision of customized user experiences, which encompass features like personalized links and community recommendations (Lazer, 2015). Bakshy et al. (2015) delved into the impact of social media, exemplified by Facebook, on user perspectives and illuminated the salient role played by individual choices. These choices include interactions within one's social circles and the deliberate consumption of specific content, both of which wield substantial influence over the extent to which individuals are exposed to divergent ideological viewpoints. Consequently, comprehending the dynamics of polarization necessitates a profound understanding of the intricate processes through which people form their opinions and perspectives, rooted in the dual forces of social influence and social selection.

A vast amount of literature on opinion dynamics tries to model the evolution of opinions mathematically and study how it affects human behavior (Bonabeau, 2002; Centola, 2018). Within the scope of this study, our primary emphasis centers on the examination of opinion dynamics as manifested within network structures. Among the well-recognized category of opinion dynamics models, a prominent subset is constituted by averaging models applied to networks. These models characterize an individual's opinion as a weighted aggregate of the opinions held by their neighbors in the network, a concept that has been extensively elaborated upon in Friedkin (1986); DeGroot (1974); Proskurnikov & Tempo (2017), and Abelson (1964).

In this paper, we seek to understand how an administrator of a social networking platform can strategically modify the network's topology while adhering to predefined budget constraints, with the overarching objective of mitigating polarization. For the rest of this paper, we assume that the underlying opinions evolve using one of the popular averaging models, Friedkin and Johnsen's opinion formulation model, which incorporates the initial opinions of individuals into the averaging process. We aim to address the scenario outlined below.

***Instance***: Consider an undirected network denoted as $G$, characterized by $V$ users (nodes) and $E$ edges. Each user maintains an immutable initial opinion. The evolution of these opinions is governed by the Friedkin-Johnsen (FJ) opinion dynamics model. Within this framework, a budget denoted as $k$, where $k > 0$, can be allocated either for distribution among the existing edges of $G$ or for adding new edges to the network.

**Problem 1.** *How can the network administrator alter the network topology within the budget $k$ to minimize polarization?*

Figure 1 shows the reduction in polarization using our proposed nonconvex relaxation on the classic Karate Club Network. While expressed or external opinions are empirically quantifiable, a fundamental limitation of the FJ model is the near impossibility of having prior knowledge of the initial opinions of all users. In many real-world scenarios, only a few users share their opinions on a social media platform about a topic, while many may prefer not to share their opinions publicly. In response to this challenge, we expand our research to address an unexplored problem setting, where the observer or administrator has access to only a subset of users' initial opinions. This is described below.

**Problem 2.** *Let $s$ represent the vector of initial opinions of users defined by $s = \begin{bmatrix} s_1^T & s_2^T \end{bmatrix}^T$, where $s_1$ denotes the vector containing the known initial opinions of users, and $s_2$ is the vector of the unknown initial opinions. How can the administrator/observer minimize polarization by altering the network topology when oblivious to $s_2$?*

**Our Main Contributions:**
- **Global Optimality:** We provide a general matrix result showing that every local minimum is a global minimum for a general class of matrix functions, $s^T M^{-k} s$, with $M \succ 0$, $s \in \mathbb{R}^n$ and an integer $k > 1$, where polarization and multiperiod polarization represent specific cases. [Theorem 1, Section 4.1]. We also extend this result to the presence of stubborn actors [Lemma 1].
- **Hardness:** We show that the minimizing polarization under integral constraints is $\mathcal{NP}$-Hard [Lemma 5].
- **Invex Relaxation for both known and partially known initial opinions :** We provide an invex (nonconvex) relaxation with guarantees of global minimum for minimizing polarization and multiperiod polarization for known initial opinions. We use projected gradient descent to solve this relaxation, notably surpassing existing state-of-the-art approaches [Section 6]. We also provide an invex (nonconvex) formulation with similar theoretical guarantees to minimize polarization when the administrator has access only to a partial set of users' initial opinions [Section 5].
- **A Novel Framework**: Our contribution centers on the introduction of a novel continuous optimization framework for minimizing single and multiperiod polarization, as well as polarization under stubborn actors. Instead of prescribing a particular method, we provide a general framework that can be employed with various continuous optimization algorithms.

**Organization:** The paper is structured as follows: Section 2 reviews the Friedkin-Johnsen model and the terminology pertinent to polarization. Section 3 discusses the prior related research. Section 4 is dedicated to a comprehensive theoretical examination of the objective function associated with

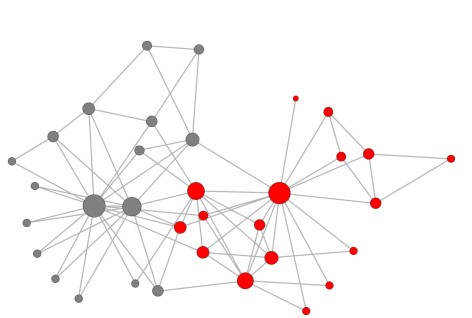
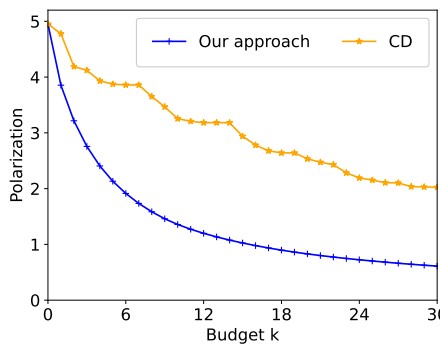

(a) Opinionated clusters in Karate Club network

(b) Change in polarization using our proposed relaxation vs. the state of art approach (CD: Coordinate Descent) across varying budget on Karate Club network.

Figure 1: Reduction in Polarization on Karate Club Network

polarization minimization. Section 5 provides nonconvex formulations designed for scenarios where the administrator/observer has partial and complete access to users' initial opinions. Finally, Section 6 presents empirical findings relevant to the problem under investigation.

**Notation:** The set of natural and real numbers is denoted by $\mathbb{N}$ and $\mathbb{R}$, respectively. For a matrix $M$, $M_{ij}$ is the entry in the $i^{th}$ row and $j^{th}$ column. The identity matrix is represented as $I$. The vectorized form of a matrix $M$ is denoted as $\text{vec}(M)$. The sets encompassing positive definite (PD) and positive semi-definite (PSD) matrices are respectively designated as $S_{++}^n$ and $S_+^n$. The Laplacian matrix of the adjacency matrix for graph G is denoted as $L$ and defined by the equation $L = D - W$, where $D$ is a diagonal matrix of (weighted) degrees associated with each node and $W$ is the weighted adjacency matrix. It is known that the graph Laplacian is a positive semi-definite matrix, and the set of Laplacian matrices $\mathcal{L}$ is a convex set. The algebraic connectivity of a given Laplacian matrix is provided by its second smallest eigenvalue, $\lambda_2$. We use $\text{Tr}$ to denote the trace of the matrix. In the context of a vector $s$, $\|s\|_1$ and $\|s\|_2$ correspond to the $\ell_1$ and $\ell_2$ norms, respectively. Furthermore, the $\ell_0$ norm signifies the count of non-zero entries within the matrix or vector.

## 2 PRELIMINARIES

### 2.1 FRIEDKIN-JOHNSEN MODEL (FJ)

FJ is a well-regarded opinion dynamics framework that considers the inherent opinions of individuals within a network (Friedkin, 1986). Within this model, we represent the immutable, initial opinions of users/actors as $s \in \mathbb{R}^n$ and the expressed opinions as $z \in \mathbb{R}^n$. Additionally, we use $w_{ij} \geq 0$ to denote the weight associated with edge $(i, j) \in E$. The fixed-point iteration at the time step $t$ governing the FJ opinion dynamics model is expressed as follows:

$$z_i^{(t)} = \frac{s_i + \sum_{j \in N(i)} w_{ij} z_j^{(t-1)}}{\sum_{j \in N(i)} w_{ij} + 1} \ . \tag{1}$$

At each discrete time step, each actor within the network adopts an expressed opinion that is proportionally influenced by the average of their own inherent opinion and the opinions held by their network neighbors. It is well-established that the FJ dynamics exhibit convergence towards an equilibrium set of opinions $z^*$ given by $z^* = (I + L)^{-1}s$ (Bindel et al., 2015). FJ model has also been studied in the presence of stubborn actors Xu et al. (2022). The fixed-point iteration governing the dynamics of node $i$ in this context is given as follows:

$$z_i^{(t)} = \frac{k_i s_i + \sum_{j \in N(i)} w_{ij} z_j^{(t-1)}}{\sum_{j \in N(i)} w_{ij} + k_i} \ . \tag{2}$$

In the equation above, the variable $k_i$, $k_i \geq 0$, signifies the degree of stubbornness for a given node. Through the iterative application of equation (2), the expressed opinion vector at equilibrium, denoted as $z^*$, is given by $(L + K)^{-1} K s$, where $K$ represents a diagonal matrix with $k_i$ as its diagonal entries. A detailed discussion of the mathematical models of social influence is provided in the supplementary section.

## 2.2 POLARIZATION UNDER FJ DYNAMICS

In this section, we formally define our problem and provide an array of definitions that are used in the literature. In the following, the notations $\bar{s}$ and $\bar{z}$ represent mean-centered initial opinions and expressed opinions, respectively. In the context of a graph $G$ with associated initial opinions, $\bar{s}$, the expressed opinions at equilibrium are determined by the expression $\bar{z} = (I + L)^{-1} \bar{s}$

**Definition 1** (Polarization)**.** *The polarization or controversy of an undirected network is defined as $\mathcal{P}(\bar{z}) = \bar{z}^T \bar{z} = \bar{s}^T (I + L)^{-2} \bar{s}$ (Chen et al., 2018; Musco et al., 2018). Polarization formalizes how close the given network is to consensus.*

**Definition 2** (Polarization-Disagreement Index)**.** *Polarization-Disagreement Index is defined as the inner product of initial and expressed opinions and is expressed as $s^T (I + L)^{-1} s$ (Chen et al., 2018).*

**Definition 3** (Polarization under stubbornness)**.** *Given an undirected network, $G$ with initial opinions, $s$, expressed opinions $z$, and the stubbornness matrix $K$ denoting the degree of stubbornness, the polarization with stubbornness is given by $\mathcal{P}(z) = s^T K (L + K)^{-1} K (L + K)^{-1} K s$.*

When $K = I$, this definition reduces to non-mean-centered polarization of expressed opinions (Xu et al., 2022).

**Definition 4** (Minimizing Polarization)**.** *When presented with a symmetric adjacency matrix $A_0$ depicting a graph along with its initial opinions, denoted as $s \in \mathbb{R}^n$, and operating under a budget constraint $k \in \mathbb{N}$, the primary aim of a network administrator is to determine a graph $G$ with a symmetric adjacency matrix $A$ that minimizes polarization. This objective can be articulated as:*

$$\begin{aligned} \underset{G}{\arg\min} \quad & \mathcal{P}(z) \\ \text{subject to} \quad & \| \operatorname{vec}(A) - \operatorname{vec}(A_0) \|_0 \leq 2k \end{aligned} \tag{3}$$

**Definition 5** (Average Conflict Risk (ACR))**.** *ACR, also referred to as expected conflict risk, is defined by taking expectation over all initial opinions. Assuming that the opinions are sampled from a uniform distribution, it evaluates to $\operatorname{Tr}((I + L)^{-2})$ (Chen et al., 2018).*

## 3 PRIOR WORK

Numerous researchers across the scientific community have been actively engaged in the study of polarization and its associated characteristics. Previous research on polarization minimization can be broadly classified into two categories: one approach centers on diminishing polarization by introducing perturbations to initial opinions, while the other attains polarization reduction through modifications to the network structure. In this work, our primary focus lies in the domain of reducing polarization by altering the network structure. For a broader review of other related research pertinent to the first category, we refer readers to the supplementary section A. We first discuss the related work pertinent to Problem 1. Musco et al. (2018) delved into the problem of determining an undirected graph topology with a prescribed edge count to minimize polarization and disagreement. Their work established the convexity of the network's Polarization-Disagreement (PD) index with respect to the Laplacian matrix $L$. Moreover, they provided proof of the existence of a graph topology with $\mathcal{O}(\frac{n}{\epsilon^2})$ edges, approximating the optimum within a factor of $(1 + \epsilon)$ through the utilization of Spielman and Srivastava's sparsification algorithm based on effective resistance (Spielman & Srivastava, 2008). Chen et al. (2018) defined polarization as the sum of squares of expressed opinions and proposed a measure called ACR (defined in 5) to minimize polarization in the presence of

an unknown opinion vector. Chitra & Musco (2020) augmented the Friedkin-Johnsen (FJ) model by establishing connections between users who share matching ideologies, aiming to minimize disagreement among users. On similar lines, Gaitonde et al. (2020) showed that the entire graph spectra of the Laplacian matrix are relevant rather than their extreme eigenvalues to maximize repeated disagreement in a network. Bhalla et al. (2023) extended the FJ model and showed how polarization increases via swaps of more agreeable opinioned edges for more disagreeable ones. Recently, Rácz & Rigobon (2023) studied how an administrator or a centralized planner can alter the network to reduce polarization. They show the nonconvexity of the polarization function and bound its value using the Cheeger constant (Chung, 1997). Furthermore, they show that the value of polarization is not monotonic by the addition of edges unless the initial opinions vector is chosen to be the eigenvector corresponding to the second smallest eigenvalue of $L$. Rácz & Rigobon (2023) explored the Fiedler difference vector approach (FD) and the coordinate descent approach (CD) as mechanisms for polarization reduction and observed that FD effectively reduces polarization without diminishing network homophily, which is defined as a tendency where similar individuals connect to each other. In the CD approach, non-edges that yield the most significant polarization reduction are iteratively added to the graph until the budget constraint is satisfied. We employ CD, FD, and ACR (defined in 5) approaches as baselines for comparative evaluation against our proposed nonconvex relaxations in Section 6. Since Problem 2 has never been dealt with before, no prior work is dedicated to it. However, related research exists in the limiting case where none of the initial opinions are observed, effectively reducing it to the problem of ACR (5) Chen et al. (2018).

## 4 THEORETICAL RESULTS

In this section, we study the global optimality of polarization. To that end, we show that it falls under a special kind of nonconvex function, namely the invex function. Invex functions can be seen as a generalization of convex functions. Hanson (1981) defined invexity as follows.

**Definition 6.** *Let $f(\theta)$ be a function defined on a set $\mathcal{C}$. Let $\eta$ be a vector-valued function defined in $\mathcal{C} \times \mathcal{C}$ such that the Frobenius inner product, $\langle \eta(\theta_1, \theta_2), \nabla f(\theta_2) \rangle$, is well defined $\forall\, \theta_1, \theta_2 \in \mathcal{C}$. Then $f(\theta)$ is a $\eta$-invex function if $f(\theta_1) - f(\theta_2) \geq \langle \eta(\theta_1, \theta_2), \nabla f(\theta_2) \rangle, \forall\, \theta_1, \theta_2 \in \mathcal{C}$.*

A function is an invex function iff it attains global minima at every stationary point Ben-Israel & Mond (1986). Next, we prove the invexity of a general class of functions. While this result can be of independent interest, we restrict our attention to minimizing polarization and related problems. Unless explicitly stated, all the formulations mentioned below work for both mean and non-mean-centered vectors. By little abuse of notation, we represent $\eta$ as a vector or matrix, depending on the specific context, in order to enhance the clarity of our presentation when the implications of such a representation are readily discernible.

**Note**: All the proofs are in the supplementary material.

**Theorem 1.** *The class of matrix functions $f(M) = s^T M^{-k}s$, with $M \succ 0$ and any integer $k > 1$ are $\eta$-invex for $\eta(\cdot, M) = M$.*

**Corollary 1.** *As a consequence of Theorem 1, the polarization function, $f(L) = s^T(I + L)^{-2}s$, is $\eta$-invex for $\eta(\cdot, L) = I + L$.*

The nonconvexity of the function $s^T M^{-2}s$ for $M \succ 0$ can be shown by restricting it to a line. For example, plot of $f(z) = s^T \begin{bmatrix} z & 0.9 \\ 0.9 & 1 \end{bmatrix}^{-2} s$ with respect to $z \in [1, 2]$ and $s = \begin{bmatrix} 1 \\ 1 \end{bmatrix}$ is visibly nonconvex (the figure is provided in the supplementary material section C). Thus, $s^T M^{-2}s$ is a nonconvex but invex function. In the following lemma, we show that the polarization remains invex even in the presence of stubborn actors.

**Lemma 1.** *Let $K$ represent the diagonal matrix of stubbornness coefficients associated with stubborn actors in the network. The polarization function $f(L) = s^T K(L+K)^{-1}K(L+K)^{-1}Ks$ is $\eta$-invex for $\eta(\cdot, L) = \frac{(L+K)}{2}$.*

**Lemma 2.** *For $L \in \mathcal{L}$, the Average Conflict Risk, $f(L) = \mathrm{Tr}(I + L)^{-2}$, is convex.*

The above result follows due to the fact that for any positive definite matrix $A$, $\mathrm{Tr}(A^{-r})$ for $r > 0$ is convex due to the positivity of trace of products of positive definite matrices (proposition 10.6.17 from Bernstein (2009)).

## 4.1 MULTIPERIOD POLARIZATION

So far, we have considered a single time period polarization. As an extension, it is natural to consider a similar objective over a prolonged time instance. We consider a $\mathcal{T}$-period polarization as an extension to one-period polarization defined in (1). In the first time period, the expressed opinions $z(\mathcal{T}(1))$ are $(I + L)^{-1}s$. These become the initial opinions for the next subsequent step, and the expressed options at the second period become $z(\mathcal{T}(2)) = (I + L)^{-2}s$. The polarization of these opinions is then added to the initial polarization. This process is repeated for $\mathcal{T} + 1$ time steps, where $\mathcal{T} \in \mathbb{N} \cup \{\infty\}$. In a multi-period setup, the objective is to minimize polarization across all time periods. By incorporating this, we get the following framework:

$$\min_{L \in \mathcal{L}} s^T[(I + L)^{-2} + (I + L)^{-4} + (I + L)^{-6} + \cdots + (I + L)^{-2\mathcal{T}-2}]s .  \quad (4)$$

**Lemma 3.** *The multiperiod polarization, i.e., the objective function given in equation equation 4, is $\eta$-invex for $\eta(\cdot, L) = I + L$.*

The following Lemma quantitatively characterizes the global minimum and helps us understand the graph structures where the global minimum is attained for multiperiod polarization.

**Lemma 4.** *The global minimum for multiperiod polarization is attained for complete graphs.*

The theoretical results provided in Theorem 1, Corollary 1, Lemmas (1, 2 and 3) imply that every local minimum is a global minimum for optimization problems such as ACR 5 and $s^T M^{-k}s$, $M \succ 0$. There exists polynomial time algorithms to solve ACR. However, the polarization function given in equations 3 is not known to be convex (Rácz & Rigobon, 2023). Moreover, Lemma 5 shows that minimizing polarization under integrality constraints is $\mathcal{NP}$-Hard. This rules out the possibility of having a polynomial time algorithm unless $\mathcal{P} = \mathcal{NP}$.

**Lemma 5.** *Let $G$ be an undirected graph with its associated graph Laplacian $L$. Let the budget $k$ denote the number of graph edits in terms of edges. For a specific choice of initial opinions vector, identifying a graph Laplacian, $L$, nearest to the given graph Laplacian, $L_0$ within a budget $k$ and having minimum adversarial polarization is $\mathcal{NP}$-hard.*

## 5 NONCONVEX RELAXATION FOR MINIMIZING POLARIZATION

While Theorem 1 and Lemma 3 establish that polarization and multiperiod polarization are invex functions, they do not readily provide a framework to solve them. Next, we develop a nonconvex relaxation framework for Problem 1 and 2 to minimize polarization. We first delve into a scenario where the observer is limited to accessing only a subset of the users' initial opinions within the network (Problem 2). The vector of initial opinions of users, denoted as $s = \begin{bmatrix} s_1^T & s_2^T \end{bmatrix}^T$, is partitioned into two components: $s_1$, comprising the known initial opinions of users, and $s_2$, representing the initial opinions that remain concealed from the observer. We assume that $s_2$ follows a distribution characterized by a zero mean and an identity covariance matrix, such as the standard Gaussian or uniform distributions. Formally, we take $\mathbb{E}(s_2) = 0$ and $\mathbb{E}(s_2 s_2^T) = I$. Let us represent $(I + L)^{-2}$ as $\begin{bmatrix} W_{11} & W_{12} \\ W_{12} & W_{22} \end{bmatrix}$, with each $W_{ij}$ being a block matrix having appropriate dimensions. For the sake of clarity, we omit the dimension details when they are evident from the context. Using the definition of polarization, we obtain:

$$f(L) = s^T(I + L)^{-2}s = \begin{bmatrix} s_1^T & s_2^T \end{bmatrix} \begin{bmatrix} W_{11} & W_{12} \\ W_{12} & W_{22} \end{bmatrix} \begin{bmatrix} s_1 \\ s_2 \end{bmatrix}$$
$$= s_1^T W_{11} s_1 + s_1^T W_{12} s_2 + s_2^T W_{12} s_1 + s_2^T W_{22} s_2$$

It is important to highlight that $f(L)$ is a random variable owing to the presence of $s_2$. Therefore, our objective is to minimize the expected polarization. Taking the expectation on both sides leads to:

$$\mathbb{E}(f(L)) = \mathbb{E}(s_1^T W_{11} s_1) + \mathbb{E}(\mathrm{Tr}(W_{22} s_2 s_2^T)) = s_1^T W_{11} s_1 + \mathrm{Tr}(W_{22})$$

While a two-step approach involving the initial minimization of $s_1^T W_{11} s_1$ followed by the minimization of $\text{Tr}(W_{22})$ might seem appealing, the budget constraint prohibits their decoupling. Our subsequent result establishes that the expected polarization $\mathbb{E}(f(L))$ is an invex function.

**Theorem 2.** *Given a vector $s \in \mathbb{R}^n$ defined as $s = \begin{bmatrix} s_1^T & s_2^T \end{bmatrix}^T$, where $s_1 \in \mathbb{R}^{n-m}$ and $s_2 \in \mathbb{R}^m$, and assuming that $s_2$ is selected from a distribution satisfying $\mathbb{E}(s_2) = 0$ and $\mathbb{E}(s_2 s_2^T) = I$, it follows that $\mathbb{E}(f(L))$ is invex.*

This result stems from the observation that the expected polarization can be expressed as a summation of invex functions. To illustrate this, we rephrase the expected polarization as $\mathbb{E}(f(L)) = a^T(I + L)^{-2}a + \sum_{i=1}^{m} b_i^T(I + L)^{-2}b_i$, where $a = \begin{bmatrix} s_1^T & 0 \end{bmatrix}^T$ and $b_i = \begin{bmatrix} 0 & e_i^T \end{bmatrix}^T$ for all $i = \{1, \cdots, m\}$, with $e_i \in \mathbb{R}^m$ denoting the standard unit vector containing a 1 at its $i$-th entry. We propose the following nonconvex relaxation for this scenario:

$$\min_{L} \quad a^T(I + L)^{-2}a + \sum_{i=1}^{m} b_i^T(I + L)^{-2}b_i$$
$$\text{subject to} \quad L \in \mathcal{L}$$
$$\| \text{vec}(L) - \text{vec}(L_0) \|_1 \leq 4k \ . \tag{5}$$

It is worth noting that the proposed nonconvex (Invex) formulation framework provides a generalization of the established Average Conflict Risk (ACR) measure (5) for the purpose of polarization minimization. Observe that we relax the nonconvex budget constraint $\ell_0$ to $\ell_1$ and express it in terms of Laplacian rather than adjacency matrix (unlike stated in equation (3)). The budget constraint has been modified to $4k$ instead of $2k$ because it affects *four* entries of the Laplacian matrix ($\{(i, j), (j, i), (i, i), (j, j)\}$).

When all initial opinions are known (Problem 1), i.e., $s = s_1$, equation 5 simplifies to:

$$\min_{L} \quad s^T(I + L)^{-2}s$$
$$\text{subject to} \quad L \in \mathcal{L}$$
$$\| \text{vec}(L) - \text{vec}(L_0) \|_1 \leq 4k \ . \tag{6}$$

This is a result of $\sum_{i=1}^{m} b_i^T(I + L)^{-2}b_i = 0$ since $b_i = 0$ (as $s_2$ is a null vector). In this paper, we aim to solve the equation 5 and equation 6. A practical limitation when solving such nonconvex formulations is that the resulting Laplacian can become dense. Even for smaller budgets, we observed that the solution tends to converge to a complete graph with smaller weights distributed across the network. To address this, we further prune the solution obtained by using a thresholding parameter $\rho$ to discard smaller weights in $L$ and set them to zero. Notice that after pruning the resultant matrix, $\hat{L}$ need not be a Laplacian. We get the optimal Laplacian $L^{proj}$ closest to $\hat{L}$ by projecting the diagonal entries: $L_{ii}^{proj} = -\sum_{j=1, j \neq i}^{n} \hat{L}_{ij}, \forall i \in \{1, \cdots, n\}$ (Sato, 2019). Only the diagonal entries need to be updated after pruning. The nonconvex relaxations mentioned above can be readily extended to address multiperiod polarization and polarization scenarios involving stubborn actors due to the invex nature of the objective functions (Lemma 1 and 3). It is worth noting that any first-order algorithm should be applicable to our framework provided in equations (5, 6) to attain global optimality. We use the projected gradient descent (PGD) algorithm to solve the equations 5, 6. In the next section, we empirically demonstrate that our relaxations lead to better minima with a few iterations of PGD.

## 6 EXPERIMENTAL RESULTS

### 6.1 FOR KNOWN INITIAL OPINIONS (PROBLEM 1)

Apart from the Coordinate Descent approach (CD) proposed by Rácz & Rigobon (2023), two other approaches to minimize polarization are to minimize $\text{Tr}((I + L)^{-2})$ (ACR defined at 5) and maximize $\lambda_2(L)$ (from Lemma 5) (Ghosh & Boyd, 2006; Wang & Van Mieghem, 2010). The heuristic approach to maximize $\lambda_2(L)$ is based on adding edges between nonadjacent vertices in the graph that have the

largest absolute difference in the entries of Fiedler vector (Chung, 1997). In this section, we compare the empirical performance of our nonconvex (invex) relaxation (equation 6) with the Coordinate Descent approach (CD) proposed by Rácz & Rigobon (2023), ACR (Tr minimization) and Fiedler Difference vector (FD) (Wang & Van Mieghem, 2010). We use the projected gradient descent method (PGD) in CVX (Diamond & Boyd, 2016; Agrawal et al., 2018) to solve our proposed nonconvex relaxation. We study the performance of our approach on real-world and synthetic networks. For synthetic networks, we consider the stochastic block models. Further experimental results on networks such as Zachary's Karate Club, Sawmill Network, The US Senate Network, Polbooks, Preferential Attachment (scale-free) graphs, and Erdös-Rényi are provided in the supplementary material.

**Stochastic Block Model:** The Stochastic Block Model (SBM) generates random graphs with inherent community structure, emphasizing node groups. In our simulation, we create two communities, each with 250 nodes. Inter-cluster and intra-cluster densities are 0.02 and 0.08, resulting in 500 nodes and 6,359 edges in the network. We distribute initial opinions in two ways: (1) assigning "-1" to one block and "+1" to the other, creating well-connected opinionated clusters (see Figure 2(a)), and (2) uniformly distributing "+1" and "-1" opinions within each block (Figure 2(b)). Across both scenarios, the invex relaxation method consistently outperforms the Coordinate Descent, $\text{Tr}$, and FD methods. We use the thresholding parameter $|\rho| = 0.0002$, step size $\alpha = 0.5$, and run PGD for 100 iterations. In the first scenario, with distinctly separated opinionated clusters, the average number of edges using our proposed nonconvex (invex) relaxation with thresholding parameter $\rho$ is 7,942. In the second scenario, with uniform opinion distribution, it is 7,616 (after thresholding).

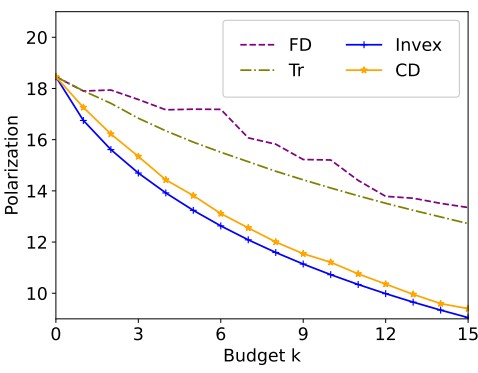 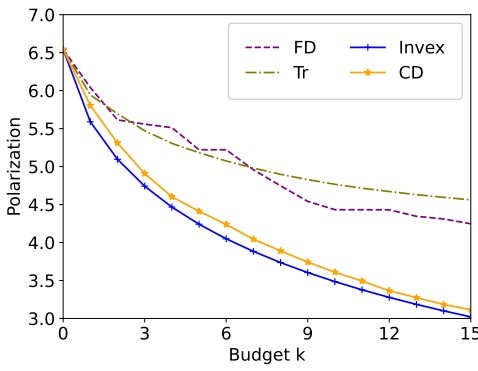

(a) Change in polarization with budget k in SBM when an initial opinion of "-1" is assigned to one community of nodes and an opinion "+1" to the other community.

(b) Change in polarization for uniformly distributed opinions within each community in SBM.

Figure 2: Reduction in Polarization on Stochastic Block Model

Our empirical analysis shows that our proposed nonconvex relaxation consistently outperforms other methods in reducing polarization. The Fiedler Difference (FD) approach primarily aims to reduce polarization by increasing algebraic connectivity, as demonstrated in Lemma 5. While raising the second smallest eigenvalue ($\lambda_2$) may cause other eigenvalues to increase as $L \in S_+^n$, this increase is insufficient for FD to achieve significant polarization reduction. In the second scenario of our construction of SBM, the FD approach seeks to maximize $\lambda_2$ by introducing additional edges within the opinionated clusters, potentially inadvertently fostering the creation of echo chambers.

**Twitter:** The Twitter dataset, originally gathered for the analysis of the Delhi legislative assembly elections debate by De et al. (2014) through hashtags such as #BJP, #AAP, #Congress, and #Polls2013, comprises an undirected network involving 548 users with a total of 3638 interactions. Initial opinions are derived from user interactions on Twitter employing sentiment analysis. Figure 3(a) illustrates the polarization variation across different budgets ($k = 1, 15, 20, 25, 30$) using our nonconvex relaxation (equation 6), CD, Trace minimization, and FD methodologies. The projected gradient descent method for equation 6 is executed for a maximum of 130 iterations across all budgets, with a step size of $\alpha = 0.5$ and a thresholding parameter $|\rho| = 0.0002$. Notably, the reduction in polarization is most pronounced when employing nonconvex relaxation (equation 6) compared to all other approaches.

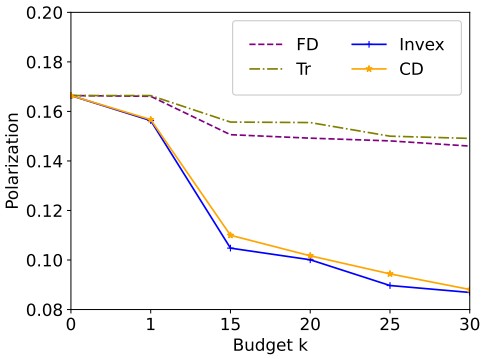
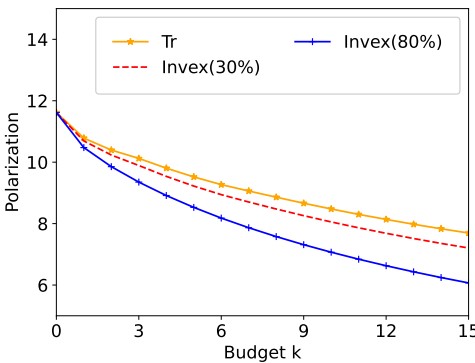

(a) Change in polarization using our proposed non-convex relaxation, CD, Trace and FD on the Twitter network.

(b) Change in polarization with budget for partially observable opinions of (30% and 80% of known initial opinions) using invex relaxation and $\mathrm{Tr}((I + L)^{-2})$

Figure 3: Reduction in Polarization on the Twitter network (Problem 1) and on SBM with partially observable initial opinions (Problem 2)

### 6.2 FOR PARTIALLY OBSERVABLE INITIAL OPINIONS (PROBLEM 2)

In this section, we study the empirical performance of our proposed invex relaxation method, as presented in equation 5, and the Average Conflict Risk (ACR) measure defined in 5. It's worth noting that equation 5 serves as a generalization of the ACR measure.

**Stochastic Block Model:** We generate an SBM model using the parameters as described in 6.1, where the unknown initial opinions of users are drawn from a uniform distribution over all vectors in $\{-1, +1\}^n$. Figure 3(b) illustrates the polarization variation with the budget, considering scenarios where the observer possesses access to $30\%$ and $80\%$ of users' initial opinions. We experimented on two partial observable percentages of initial opinion. It is evident that our proposed nonconvex (invex) relaxation consistently outperforms the Average Conflict Risk (ACR) measure and is equal to its value $\mathrm{Tr}(I + L)^{-2}$ only when the observer has no knowledge of any users' opinions.

**Interpretation in social context:** Based on empirical observations, our optimization approaches presented in equations equation 5 and equation 6 effectively minimize polarization by introducing additional edges among users with polarized opinions. This aligns with findings from previous research, including Chitra & Musco (2020); Kahneman (2011); Rácz & Rigobon (2023). Utilizing continuous relaxation techniques as demonstrated in equation 5 and equation 6, we can identify significant interactions within a social network, typically represented by edges with high weights that play a pivotal role in the minimization of polarization. Armed with this insight, a network administrator can offer link recommendations and promote exposure to diverse content among network users. This strategic approach helps prevent the reinforcement of like-minded opinions, ultimately contributing to the reduction of polarization within the network.

**Conclusion and Future Directions:** This paper addresses polarization mitigation by altering network topology in two scenarios: when initial opinions are known and when the observer has partial knowledge of the opinions. We introduce a novel nonconvex relaxation framework for known opinions and demonstrate the projected gradient descent's efficacy in polarization minimization. We extend this to scenarios with incomplete knowledge of initial opinions, proposing a novel nonconvex formulation that generalizes the ACR (trace minimization) approach. Continuous relaxation techniques, as shown in equation 5 and equation 6, identify pivotal interactions that can be leveraged to provide link recommendations and diversify content exposure to mitigate polarization.

Existing scalability studies primarily focus on the computation of the polarization, denoted as $s^T (I + L)^{-2} s$ (Xu et al., 2021). In the future, it might be of significant interest to explore the applicability of analogous concepts, in conjunction with our findings, to minimize polarization for larger network configurations.

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
