## A    FURTHER RELATED WORK

In this section, we present additional references that pertain to research on polarization using FJ dynamics. Guerra et al. (2013) identified a characteristic of polarized networks, namely, a lower concentration of high-degree nodes in the vicinity of boundaries separating distinct communities. Zhu et al. (2021) provided a scalable greedy algorithm for optimizing the polarization-disagreement index for a given graph by adding a set of edges. They show that the index is monotonic with respect to the addition of edges, and despite the function being non-submodular, they provided a bounded approximation ratio. Chen & Rácz (2021) explored the amplification of disagreement and polarization through perturbations in the initial opinions held by network nodes. Bhalla et al. (2021) scrutinized the dependence of polarization on localized edge dynamics, revealing that the introduction of an edge between closely affiliated like-minded users leads to an increase in polarization. Matakos et al. (2017) provided greedy heuristics to minimize polarization by perturbing initial opinions.

## B    PROOFS OF THEOREMS AND LEMMAS

### B.1    PROOF OF THEOREM 1

*Proof.* We first compute the gradient of the function. Let $X = M^k$. Then $f(M) = s^T(X)^{-1}s$. It is known that:

$$\frac{\partial X^{-1}}{\partial m_{ij}} = -X^{-1}\frac{\partial X}{\partial m_{ij}}X^{-1} \ . \tag{7}$$

By product rule,

$$\frac{\partial M^k}{\partial m_{ij}} = J^{ij}M^{k-1} + MJ^{ij}M^{k-2} + \cdots + M^{k-1}J^{ij} \ , \tag{8}$$

where $J^{ij}$ is the matrix with 1 at $(i,j)^{th}$ entry and zero else where. By substituting eq equation 8 in equation 7, we get:

$$\frac{\partial s^T X^{-1}s}{\partial m_{ij}} = -s^T M^{-k}J^{ij}M^{-1}s - s^T M^{-(k-1)}J^{ij}M^{-2}s - \cdots - s^T M^{-1}J^{ij}M^{-k}s \ .$$

Considering $M^{-l}$ as $A$ and $M^{-q}$ as $B$ and using identity that $s^T AJ^{ij}Bs = (A^T ss^T B^T)_{ij}$ (eq (454) from Petersen et al. (2008)) we get

$$\frac{\partial s^T X^{-1}s}{\partial m_{ij}} = -(M^{-k}ss^T M^{-1})_{ij} - (M^{-(k-1)}ss^T M^{-2})_{ij} - \cdots - (M^{-1}ss^T M^{-k})_{ij} \ .$$

This implies:

$$\frac{\partial s^T M^{-k}s}{\partial M} = -M^{-k}ss^T M^{-1} - M^{-(k-1)}ss^T M^{-2} - \cdots - M^{-1}ss^T M^{-k} \ . \tag{9}$$

Equation equation 9 represents the gradient of the function $s^T M^{-k}s$ with respect to $M$. Let $M$, $N \in S_{++}^n$. To show invexity for function a $f$, we need to show that there exists an $\eta(N, M)$ such that

$$f(N) - f(M) \geq \langle \eta(N, M), \nabla f(M)\rangle \ .$$

In our case, this implies that we need to show the existence of $\eta(N, M)$ such that

$$s^T N^{-k}s - s^T M^{-k}s \geq \left\langle \eta(N, M), \frac{\partial s^T M^{-k}s}{\partial M}\right\rangle \ .$$

After substituting for the gradient, we get

$$s^T N^{-k} s - s^T M^{-k} s \geq - \langle \eta(N, M), M^{-k} ss^T M^{-1} \rangle - \cdots - \langle \eta(N, M), M^{-1} ss^T M^{-k} \rangle .$$

With little algebraic manipulation, we can write

$$s^T N^{-k} s - s^T M^{-k} s \geq - \text{Tr}(\eta(N, M)^T M^{-k} ss^T M^{-1}) - \cdots - \text{Tr}(\eta(N, M)^T M^{-1} ss^T M^{-k}) .$$

The right-hand side of the above expression can be expressed as

$$- \sum_{i=0}^{k-1} \text{Tr}(s^T M^{-(i+1)} \eta(N, M)^T M^{-(k-i)} s) .$$

By choosing $\eta(N, M) = M$, we get

$$s^T N^{-k} s - s^T M^{-k} s \geq - \sum_{i=0}^{k-1} \text{Tr}(s^T M^{-k} s) ,$$

which implies

$$s^T N^{-k} s + \sum_{i=0}^{k-2} s^T M^{-k} s \geq 0 .$$

The above result follows because of the positive definiteness of $N$ and $M$. To complete the proof, we also need to show that if $\nabla f(M) = 0$, then $f(N) \geq f(M), \forall N$, i.e., the stationary point is indeed the global minimum of the function. By equating the gradient to zero, we get

$$-M^{-k} ss^T M^{-1} = M^{-(k-1)} ss^T M^{-2} + \cdots + M^{-1} ss^T M^{-k} .$$

Right multiplication with $M$ gives us

$$-M^{-k} ss^T = M^{-(k-1)} ss^T M^{-1} + \cdots + M^{-1} ss^T M^{-(k-1)} ,$$

which implies

$$- \text{Tr}(M^{-k} ss^T) = \text{Tr}(M^{-(k-1)} ss^T M^{-1}) + \cdots + \text{Tr}(M^{-1} ss^T M^{-(k-1)}) .$$

It follows that

$$- \text{Tr}(s^T M^{-k} s) = \text{Tr}(s^T M^{-k} s) + \cdots + \text{Tr}(s^T M^{-k} s) ,$$

and thus

$$s^T M^{-k} s = 0 .$$

The above equation shows that this class of functions does not have any stationary point.

$$\square$$

## B.2    Proof for Lemma 1

*Proof.* Let $x = s^T K$. Then $f(L) = x^T (L + K)^{-1} K (L + K)^{-1} x$. The gradient of the function is given by

$$\nabla f(L) = -(L + K)^{-1} xx^T (L + K)^{-1} K (L + K)^{-1} - (L + K)^{-1} K (L + K)^{-1} xx^T (L + K)^{-1} .$$

Let $L_1, L_2 \in S_+^n$. To show invexity for function $f$, we need to show that there exists an $\eta(L_1, L_2)$ such that

$$f(L_1) - f(L_2) \geq \langle \eta(L_1, L_2), \nabla f(L_2) \rangle .$$

For our problem, this means that we need to show

$$x^T (L_1 + K)^{-1} K (L_1 + K)^{-1} x - x^T (L_2 + K)^{-1} K (L_2 + K)^{-1} x \geq$$
$$- \langle \eta(L_1, L_2), (L_2 + K)^{-1} xx^T (L_2 + K)^{-1} K (L_2 + K)^{-1} \rangle$$
$$- \langle \eta(L_1, L_2), (L_2 + K)^{-1} K (L_2 + K)^{-1} xx^T (L_2 + K)^{-1} \rangle$$
$$= - \text{Tr}(\eta(L_1, L_2)^T (L_2 + K)^{-1} xx^T (L_2 + K)^{-1} K (L_2 + K)^{-1})$$
$$- \text{Tr}(\eta(L_1, L_2)^T (L_2 + K)^{-1} K (L_2 + K)^{-1} xx^T (L_2 + K)^{-1})$$
$$= - \text{Tr}(x^T (L_2 + K)^{-1} K (L_2 + K)^{-1} \eta(L_1, L_2)^T (L_2 + K)^{-1} x)$$
$$- \text{Tr}(x^T (L_2 + K)^{-1} \eta(L_1, L_2)^T (L_2 + K)^{-1} K (L_2 + K)^{-1} x)$$

for a particular choice of $\eta(L_1, L_2)$. By choosing $\eta(L_1, L_2) = \frac{L_2 + K}{2}$, we get

$$x^T(L_1 + K)^{-1}K(L_1 + K)^{-1}x - x^T(L_2 + K)^{-1}K(L_2 + K)^{-1}x \geq$$
$$- \operatorname{Tr}(x^T(L_2 + K)^{-1}K(L_2 + K)^{-1}x) .$$

As $(L_1 + K)^{-1}$ is a symmetric positive definite matrix, the matrix obtained by left multiplying it with a positive diagonal matrix is the same as right multiplying it with the same diagonal matrix and is positive definite. Thus

$$x^T(L_1 + K)^{-1}K(L_1 + K)^{-1}x = x^T(L_1 + K)^{-1}K^{\frac{1}{2}}K^{\frac{1}{2}}(L_1 + K)^{-1}x \geq 0 .$$

By following similar computation as shown in Theorem 1, it can be observed that the function has no stationary points and is $\eta$-invex for $\eta(\cdot, L) = \frac{(L+K)}{2}$. $\square$

### B.3 PROOF FOR LEMMA 3

*Proof.* From Theorem 1 we know that the class of functions $f(I + L) = s^T(I + L)^{-k}s$ are $\eta$-invex for $\eta(\cdot, L) = I + L$. Using the linearity of trace and partial derivative operators and following the similar computation as shown in Theorem (1), we can conclude that $\sum_{i=1}^{T} s^T(I + L)^{-2i}s$ is $\eta$-invex for $\eta(\cdot, L) = I + L$. $\square$

### B.4 PROOF OF LEMMA 4

*Proof.* Recall that the Laplacian spectrum of the complete graph has an eigenvalue 0 with multiplicity 1 and an eigenvalue of $n$ with multiplicity $n - 1$. When the opinions are mean-centered opinion vectors $s$ (such that $s^T 1 = 0$), the expressed opinions are given by $z = (I + L(K_n))^{-1}s = \frac{s}{n+1}$. The polarization of expressed opinions in the first time period is $z^T z = \|z\|^2 = \frac{\|s\|_2}{(n+1)^2}$. The $\mathcal{T}$-period polarization for the complete graph is

$$\frac{\|s\|_2}{(n+1)^2} + \frac{\|s\|_2}{(n+1)^4} + \cdots + \frac{\|s\|_2}{(n+1)^{2\mathcal{T}}} .$$

As each element in the above summation is the lower bound for the corresponding terms from the repeated polarization function, the global minimum for (4) is attained for $K_n$. $\square$

### B.5 PROOF FOR LEMMA 5

*Proof.* Consider the following two optimization problems:

$$\underset{L}{arg min} \quad \underset{s \in R^n, s \perp 1, \|s\|_2^2 \leq 1}{\max} \quad s^T(I + L)^{-2}s$$
$$\text{subject to} \qquad L \in \mathcal{L}, \tag{10}$$
$$L_{ij} = \{-1, 0\}, \; for \; i \neq j$$
$$\| \operatorname{vec}(L) - \operatorname{vec}(L_0) \|_0 \leq 4k ,$$

and

$$\underset{L}{\max} \qquad \lambda_2(L) \tag{11}$$
$$\text{subject to} \quad L \in \mathcal{L},$$
$$L_{ij} = \{-1, 0\}, \; for \; i \neq j$$
$$\| \operatorname{vec}(L) - \operatorname{vec}(L_0) \|_0 \leq 4k .$$

Mosk-Aoyama (2008), showed that finding a set of edges within a specified budget to add to the graph so that the algebraic connectivity of the augmented graph is maximized is NP-hard. By Courant-Fischer theorem (Golub & Van Loan, 2013), we can observe that the inner maximization problem in (10) takes the maximum value of $\frac{1}{(1+\lambda_2(L))^2}$, when $s$, the mean-centered initial opinion vector, is the second smallest eigenvector of $L$. Thus for the outer minimization problem, we need an $L$

obtained from $L_0$ by adding $k$ edges and with maximum $\lambda_2$. The graph associated with the Laplacian matrix returned by equation (10) is the same as the solution of equation (11). Thus, the computational hardness of minimizing polarization given in equation (10) is at least that of maximizing algebraic connectivity within the budget $k$.

$\square$

### B.6 Proof of Theorem 2

*Proof.* In the following we represent $(I + L)^{-2}$ as $\begin{bmatrix} W_{11} & W_{12} \\ W_{12} & W_{22} \end{bmatrix}$, with each $W_{ij}$ being a block matrix having appropriate dimensions. For the sake of clarity, we omit the dimension details when they are evident from the context. For a given set of initial opinions vector $s = \begin{bmatrix} s_1^T & s_2^T \end{bmatrix}^T$, the polarization function can be expressed as follows:

$$f(L) = s^T (I + L)^{-2} s = \begin{bmatrix} s_1^T & s_2^T \end{bmatrix} \begin{bmatrix} W_{11} & W_{12} \\ W_{12} & W_{22} \end{bmatrix} \begin{bmatrix} s_1 \\ s_2 \end{bmatrix}$$
$$= s_1^T W_{11} s_1 + s_1^T W_{12} s_2 + s_2^T W_{12} s_1 + s_2^T W_{22} s_2$$

On taking expectation with respect to the vector of unknowns $s_2$ we get

$$\mathbb{E}(f(L)) = s_1^T W_{11} s_1 + \mathrm{Tr}(W_{22})$$

Observe that the above equation can be rewritten as

$$\mathbb{E}(f(L)) = a^T (I + L)^{-2} a + \sum_{i=1}^{m} b_i^T (I + L)^{-2} b_i \tag{12}$$

where $a = \begin{bmatrix} s_1^T & 0 \end{bmatrix}^T$ and $b_i = \begin{bmatrix} 0 & e_i^T \end{bmatrix}$ for all $i = \{1, \cdots, m\}$, with $e_i \in \mathbb{R}^m$ denoting the standard unit vector containing a 1 at its $i$-th entry. Notice that $a^T (I + L)^{-2} a$ and $\sum_{i=1}^{m} b_i^T (I + L)^{-2} b_i$ are $\eta$-invex. Using the linearity of trace and partial derivative operators and following the similar computation as shown in Theorem (1), we can conclude that $\mathbb{E}(f(L)) = a^T (I+L)^{-2} a + \sum_{i=1}^{m} b_i^T (I+L)^{-2} b_i$ is $\eta$-invex for $\eta(\cdot, L) = I + L$. $\square$

## C Example to demonstrate the nonconvexity of the function $s^T M^{-2} s$

Here, we provide a visual depiction illustrating the nonconvex nature of the function $s^T M^{-2} s$, $M \in S_{++}^n$. In Figure 4, we plot the function $f(z) = s^T \begin{bmatrix} z & 0.9 \\ 0.9 & 1 \end{bmatrix}^{-2} s$ with respect to $z \in \begin{bmatrix} 1 & 2 \end{bmatrix}$ and $s = \begin{bmatrix} 1 \\ 1 \end{bmatrix}$. Notice that this function is nonconvex.

## D Additional Empirical Results for known initial opinions (Problem 1)

In this section, we provide additional empirical results on various networks for minimizing polarization for known initial opinions (experiments are run using CVX solver (Diamond & Boyd, 2016; Agrawal et al., 2018; Grant & Boyd, 2014; 2008)).

**Multi-period polarization**: Note that the Laplacian that minimizes single-period polarization also minimizes multi-period polarization. In this section, we provide experimental details on single-period polarization (equation 5 and equation 6) and the performance of various approaches to minimize multi-period polarization equation 4 can directly be inferred from their performance in minimizing single-period polarization.

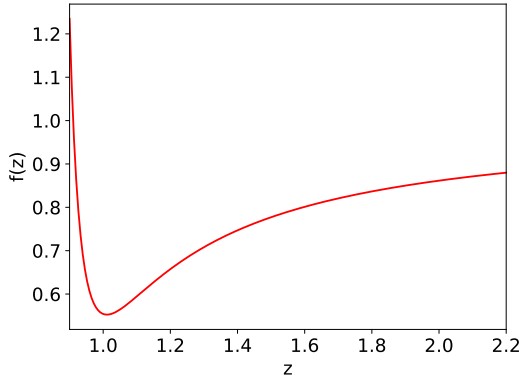

Figure 4: Nonconvexity of the function $s^T M^{-2} s$

**Karate Club network:** This network represents a social conflict between an instructor and an administrator within a karate club, as documented by Zachary (1977). It is an undirected network comprising 34 nodes and 78 edges, where each node corresponds to a club member, and edges signify connections between members. Figure 1(a) illustrates the division of club members into two opinionated clusters due to the conflict. We attribute an initial opinion of "+1" or "-1" to each opinionated cluster.

In Figure 5, we present the polarization variations across different budget allocations for our invex relaxation model equation 6, the Coordinate Descent (CD) method, Tr minimization, and the Fiedler Difference (FD) approach. It is evident that the invex relaxation model consistently outperforms CD and other methods in terms of polarization reduction. FD reduces polarization by adding a single edge, resulting in the sparsest graph configuration. For our invex relaxation approach, by utilizing the thresholding parameter $|\rho| = 0.0002$, with 100 iterations of PGD, and employing a step size of $\alpha = 0.5$, the average number of edges across different budgets amounts to 184.

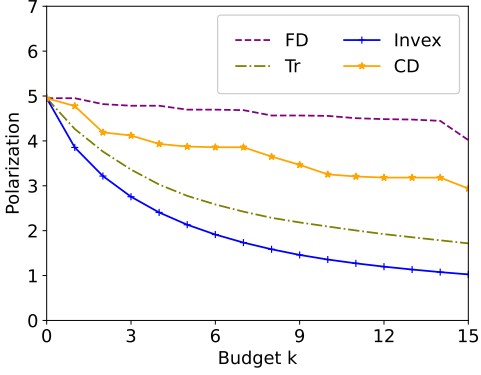

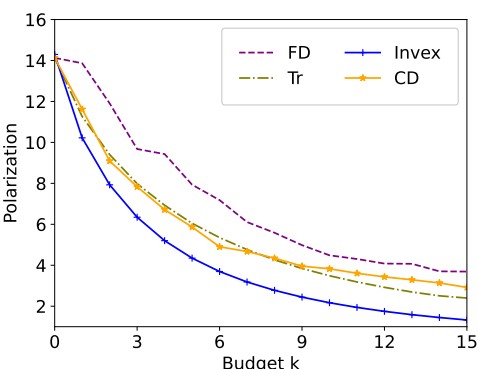

(a) Change in polarization with a budget on Karate Club network. Our nonconvex relaxation considerably reduces polarization compared to all other approaches.

(b) Change in Polarization on Sawmill Strike Network. Invex relaxation produces the best reduction in polarization compared to CD.

Figure 5: Polarization on Karate and Sawmill Networks

**Sawmill Strike network:** This network represents employees working at a sawmill during a period of strike. It is an undirected network comprising 24 nodes and 76 edges. The strike's prolonged duration was believed to be due to ineffective communication between two distinct groups of employees within the network. The network was initially analyzed in Michael (1997) to identify leaders during the strike. In this study, we leverage this network to identify potential edges that could

minimize polarization. We attribute an initial opinion of "+1" to one group and "-1" to another group of nodes.

Figure 5 (b) depicts the variation in polarization as the budget increases. Notably, our invex relaxation approach consistently achieves the most substantial reduction in polarization across different budget allocations when compared to the Coordinate Descent (CD) method. For our invex relaxation method, employing $|\rho| = 0.0002$ and with a step size of $\alpha = 0.5$, the average number of added edges amounts to 190.

**The US Senate:** This network captures the co-sponsorship of bills among US senators during session 114, as documented by Neal (2022). In this representation, each senator assumes the role of either a sponsor or co-sponsor of a bill, and edges between senators signify their joint co-sponsorship of a bill during that session. Recent studies, such as those by Hohmann et al. (2023) and Neal (2020), have explored the relevance of such co-sponsorship networks in the context of polarization. This particular network encompasses a total of 102 nodes, with 46 Democrats, 54 Republicans, and 2 Independents, interconnected by 1832 edges. We assign an initial opinion of "+1" to Democrats, "−1" to Republicans, and "0" to Independents.

Figure 6 visually presents the polarization reduction achieved using our proposed invex relaxation (equation 6), comparing it to the Coordinate Descent ((Rácz & Rigobon, 2023)), the Tr minimization, and the Fiedler Difference (FD) approaches. In our computational experiments, we ran projected gradient descent for 100 iterations, employing a step size of $\alpha = 0.2$ and setting $|\rho| = 0.0002$. The average number of edges added across all budgets amounts to 2436. The results, as depicted, demonstrate that our invex relaxation (equation 6) significantly outperforms all existing approaches in terms of minimizing polarization.

**Polbooks:** This network comprises books related to US politics and was compiled during the 2004 presidential election, as documented by Rossi & Ahmed (2015). Interactions within the network reflect instances where customers on the Amazon platform frequently purchased these books together. The books are categorized based on their political leanings, falling into three categories: Liberal, Conservative, or Neutral. Specifically, there are a total of 43 books classified as Liberal, 49 as Conservative, and 13 as Neutral. We assign an initial opinion of "+1" to Liberal, "−1" to Conservative, and "0" to Neutral. Figure 6(a) illustrates the variation in polarization across different budgets. The projected gradient descent for invex relaxation is executed for a maximum of 100 iterations, utilizing a step size of $\alpha = 0.2$ and $|\rho| = 0.0002$.

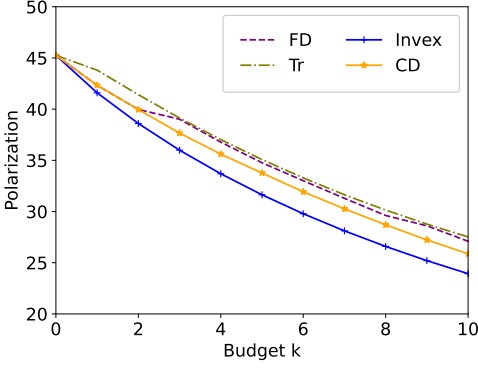

(a) Change in polarization with budget using invex relaxation, CD, Tr and FD on Polbooks

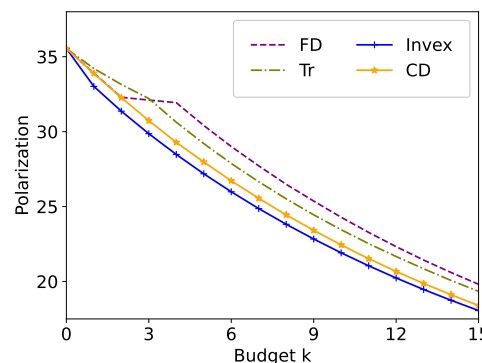

(b) Reduction in polarization with budget using invex relaxation, CD, Tr, and FD approaches on the US Senate Network.

Figure 6: Reduction in Polarization on Polbooks and US Senate networks

**Preferential Attachment (Scale Free) Network:** Preferential Attachment (PA) describes a mechanism of graph evolution where higher-degree nodes have a greater probability of receiving new neighbors. It is designed to model the power law behavior (Faloutsos et al., 1999). For our analysis, an incoming vertex connects to at most four other existing vertices in the graph. The resultant PA

network has 200 nodes and 768 edges. Nodes in the network are assigned an initial opinion of "+1" and "−1" uniformly at random.

Figure 7(a) visually illustrates the reduction in polarization across budgets ranging from $k = 1$ to $k = 15$. Notably, our invex relaxation method (equation equation 6) consistently achieves the lowest polarization compared to other approaches. In our computational experiments, we executed projected gradient descent for up to 100 iterations, employing a step size of $\alpha = 0.8$ and $|\rho| = 0.0002$. On average, after applying the thresholding parameter $\rho$, the invex relaxation approach added $1,410$ edges across all budgets.

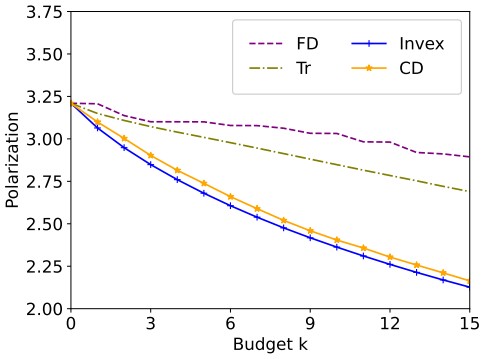
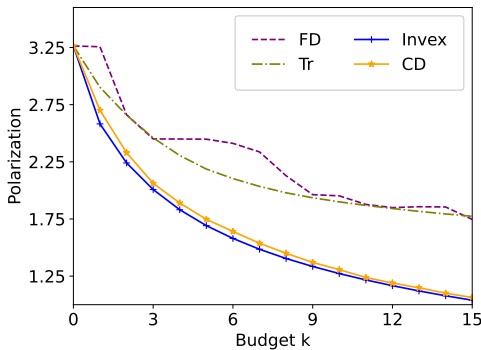

(a) Reduction in polarization with varying budgets using invex relaxation, CD, Tr, and FD approaches on Preferential Attachment network.

(b) Reduction in polarization with varying budgets using invex relaxation, CD, Tr, and FD approaches on Erdös-Rényi Graph.

Figure 7: Reduction in Polarization on Preferential Attachment and Erdös-Rényi graphs

**Erdös-Rényi:**  In this model, each pair of vertices are connected independently with a probability $p$ (Erdos & Renyi, 1960). We construct an Erdös-Rényi graph with 100 vertices and $p = 0.1$. Nodes in the network are assigned an initial opinion of "+1" and "−1" uniformly at random. The step size for invex relaxation equation 6 is set to $\alpha = 0.8$. The projected gradient descent on equation 6 is run for 100 iterations with thresholding parameter $\rho = 0.0002$. The change in polarization is depicted in Figure 7(b).

# E  ADDITIONAL EXPERIMENTATION FOR PARTIALLY OBSERVABLE INITIAL OPINIONS:

Here, we present an additional baseline for comparison with the findings depicted in Figure 3(b). We employ the Coordinate Descent (CD) approach ((Rácz & Rigobon, 2023)), which necessitates complete knowledge of initial opinions. To facilitate our experimentation with CD, we estimate unknown opinions using mean imputation, specifically setting $s_2 = \text{mean}(s_1)$. The corresponding outcome is illustrated in Figure 8. It is evident that CD outperforms Trace when it has access to larger percentage of initial opinions.

# F  INFLUENCE MODELS

In this section, we will review some of the most commonly used social influence models. We assume a real-valued, one-dimensional, continuous opinion space. In particular, we focus on linear continuous opinion models such as the DeGroot (1974) and Friedkin (1986). For simplicity, we choose the opinions to be scalar. Mathematically, they can also be a vector quantity representing an individual stance over various social phenomena.

## F.1  FRENCH-DEGROOT MODEL

French Jr (1956) proposed one of the first mathematical models for opinion formation and a group's collective behavior. Along these lines, DeGroot (1974) generalized this method and named it "iterative

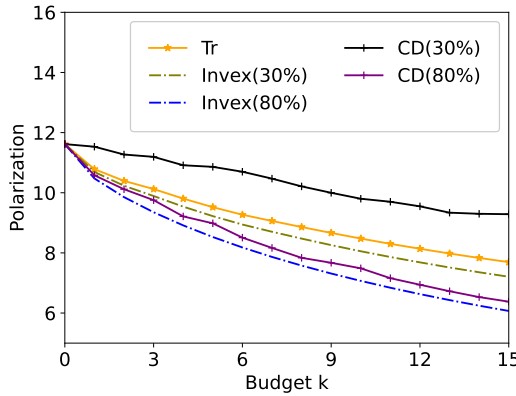

Figure 8: Change in polarization with budget for partially observable opinions of (30% and 80% of known initial opinions) using invex relaxation, CD (with mean imputation, i.e., $s_2 = \text{mean}(s_1)$) and $\text{Tr}((I + L)^{-2})$

opinion pooling". This model describes a social learning process of opinion formation based on observing other individuals in the network. It formalizes when and how quickly several actors can reach a consensus of beliefs. In this model, the individuals' opinion is modeled as the harmonic average of the opinions of their neighbors in the network. Mathematically, the opinion update rule for estimates is given by the following equation:

$$z_i^{(t)} = \frac{1}{deg(i)} \sum_{j \in N(i)} w_{ij} z_j^{(t-1)} \; . \tag{13}$$

Here $w_{ij}$ represents the weight of $j$'s opinion on $i$, and the opinion of $i$ at time $t$ is written as $z_i^{(t)}$. The open neighborhood of vertex $i$ in $G$ is denoted by $N(i)$. The DeGroot model always converges to consensus when the graph is connected.

### F.2 FRIEDKIN-JOHNSEN MODEL (FJ)

Friedkin and Johnsen generalized the DeGroot model by taking into account prejudice or initial opinions of individuals in the network (Friedkin, 1986). Let $s \in R^n$ represent the initial opinions of actors in the network. In the opinion dynamics process, this vector is assumed to be immutable. Let $z \in R^n$ denote the expressed opinions. Let $w_{ij} \geq 0$ denote the weight on edge $(i, j) \in E$. Fixed point iteration of the FJ opinion dynamics model is then given as

$$z_i^{(t)} = \frac{s_i + \sum_{j \in N(i)} w_{ij} z_j^{(t-1)}}{\sum_{j \in N(i)} w_{ij} + 1} \; . \tag{14}$$

At each time step, every actor adopts an expressed opinion that is proportional to the average of its own initial opinion and the opinion of its neighbors. It is well known that the above-defined FJ dynamics converge to an equilibrium set of opinions $z^*$ (Bindel et al., 2015) given by

$$z^* = (I + L)^{-1} s \; . \tag{15}$$

In the above expression, $I$ is an Identity matrix, and $L$ is the combinatorial Laplacian of $G$ given by $D - W$. Note that $(I + L)$ is a positive definite matrix, and hence the inverse exists. From the equation (15), we can also observe that the expressed opinions are a contraction of initial opinions, i.e., $z_i$ is a convex combination of initial opinions of all nodes, including node $i$ in the network. Consensus is not guaranteed in FJ dynamics. Bindel et al. (2015) used this to quantify the price for not reaching the consensus. They show that updating $z_i$ as given in equation (14) is the same as minimizing the following quadratic function:

$$\min_{z_i} \quad (z_i - s_i)^2 + \sum_{j \in N(i)} w_{ij}(z_i - z_j)^2 \ .$$

The term $(z_i - s_i)^2$ is the stress incurred at node $i$ due to the difference between its initial and expressed opinions (also known as internal conflict) and the second term, $\sum_{j \in N(i)} w_{ij}(z_i - z_j)^2$, as the external conflict incurred due to the difference between the expressed opinions of the node $i$ and its neighbors.

### F.3 IN-HOMOGENOUS STUBBORNNESS IN FJ MODEL

The stubbornness of actors/nodes in the network is defined as the degree of resilience to change from their initial opinions. Recently Xu et al. (2022) studied the Friedkin-Johnsen model in the presence of in-homogeneous stubbornness. The fixed point iteration of a node $i$ on a graph $G$ where every node has a certain degree of stubbornness to their initial opinions is then given as

$$z_i^{(t)} = \frac{k_i s_i + \sum_{j \in N(i)} w_{ij} z_j^{(t-1)}}{\sum_{j \in N(i)} w_{ij} + k_i} \ . \tag{16}$$

In the above equation, $k_i$ denotes the the degree of stubbornness and $k_i \geq 0$. By iterating the above equation, the expressed opinion vector at equilibrium $z^*$ is given as

$$z^* = (L + K)^{-1} K s \ , \tag{17}$$

where $K$ is a diagonal matrix with the degree of the stubbornness of each node in the network as its diagonal entries. From (17), we see that if the initial opinions of all nodes are perturbed by a constant $c$, the expressed opinions are changed to $z^* + c$.

## G   INVEXITY OF POLARIZATION WITH PARTIALLY KNOWN OPINIONS

In this section, we revisit the results presented in Section 5, specifically focusing on Theorem 2, but with less restrictive assumptions concerning the distribution of the unknown initial opinions $s_2$. While we maintain the assumption of zero mean for these opinions, we now allow for a more general covariance matrix. Our objective is to establish the following result:

**Theorem 3.** *Given a vector $s \in \mathbb{R}^n$ defined as $s = \begin{bmatrix} s_1^T & s_2^T \end{bmatrix}^T$, where $s_1 \in \mathbb{R}^{n-m}$ and $s_2 \in \mathbb{R}^m$, and assuming that $s_2$ is selected from a distribution satisfying $\mathbb{E}(s_2) = 0$ and $\mathbb{E}(s_2 s_2^T) = \Sigma$, it follows that $\mathbb{E}(f(L))$ is invex.*

*Proof.* Borrowing the notations from the Proof of Theorem 2, we represent $(I + L)^{-2}$ as $\begin{bmatrix} W_{11} & W_{12} \\ W_{12} & W_{22} \end{bmatrix}$, where each $W_{ij}$ is a block matrix with appropriate dimensions. For clarity, we omit dimension details when evident. For a given initial opinions vector $s = \begin{bmatrix} s_1^T & s_2^T \end{bmatrix}^T$, the polarization function is expressed as:

$$\begin{aligned} f(L) = s^T (I + L)^{-2} s &= \begin{bmatrix} s_1^T & s_2^T \end{bmatrix} \begin{bmatrix} W_{11} & W_{12} \\ W_{12} & W_{22} \end{bmatrix} \begin{bmatrix} s_1 \\ s_2 \end{bmatrix} \\ &= s_1^T W_{11} s_1 + s_1^T W_{12} s_2 + s_2^T W_{12} s_1 + s_2^T W_{22} s_2 \end{aligned}$$

Taking the expectation with respect to the vector of unknowns $s_2$, we obtain:

$$\mathbb{E}(f(L)) = s_1^T W_{11} s_1 + \mathbb{E}(s_2^T W_{22} s_2) \tag{18}$$

$$= s_1^T W_{11} s_1 + \mathbb{E}(\text{Tr}(W_{22} s_2 s_2^T)) \tag{19}$$

$$= s_1^T W_{11} s_1 + \text{Tr}(W_{22} \mathbb{E}(s_2 s_2^T)) \tag{20}$$

$$= s_1^T W_{11} s_1 + \text{Tr}(W_{22} \Sigma), \tag{21}$$

where equation equation 20 follows due to the linearity of the trace function.

Since covariance matrix $\Sigma$ is a positive semidefinite matrix, it has a unique square root, i.e., $\Sigma = BB^T$ for a symmetric square matrix $B$. Using this property along with the cyclicity property of trace, we rewrite equation 21 as below:

$$\mathbb{E}(f(L)) = s_1^T W_{11} s_1 + \text{Tr}(B W_{22} B) \tag{22}$$

If we represent $B = \begin{bmatrix} b_1 & b_2 & \cdots & b_m \end{bmatrix}$ for vectors $b_i \in \mathbb{R}^m, \forall i = \{1, \cdots, m\}$, then equation equation 22 can be expressed as:

$$\mathbb{E}(f(L)) = s_1^T W_{11} s_1 + \sum_{i=1}^{m} b_i^T W_{22} b_i, \tag{23}$$

which can be further rewritten as

$$\mathbb{E}(f(L)) = a^T (I + L)^{-2} a + \sum_{i=1}^{m} \bar{b}_i^T (I + L)^{-2} \bar{b}_i, \tag{24}$$

where $a = \begin{bmatrix} s_1^T & 0 \end{bmatrix}^T$ and $\bar{b}_i = \begin{bmatrix} 0 & b_i^T \end{bmatrix}^T$ for all $i = \{1, \cdots, m\}$. Recall that $a^T (I + L)^{-2} a$ and $\sum_{i=1}^{m} \bar{b}_i^T (I + L)^{-2} \bar{b}_i$ are $\eta$-invex. Using the linearity of trace and partial derivative operators and following the similar computation as shown in Theorem (1), we can conclude that $\mathbb{E}(f(L)) = a^T (I + L)^{-2} a + \sum_{i=1}^{m} \bar{b}_i^T (I + L)^{-2} \bar{b}_i$ is $\eta$-invex for $\eta(\cdot, L) = I + L$. $\qquad \square$