# OpenReview forum: "An Invex Relaxation Approach for Minimizing Polarization from Fully and Partially Observed Initial Opinions"
_ICLR.cc/2024/Conference — Submitted to ICLR 2024_

### Official Review · Reviewer_f5dH · 2023-10-19

**Soundness:** 2 fair
**Presentation:** 3 good
**Contribution:** 2 fair
**Rating:** 3
**Confidence:** 2

**Summary:**

This paper proposed an invex relaxation approach for minimizing polarization over a network. It is proved in Section 4 that many types of polarization all fall into the invex function class, whose local minimum is a global minimum. Then this paper proposes to use projected gradient descent to solve a relaxed problem.

**Strengths:**

1. The paper is well written.
2. Invexity is provably identified for many types of polarization. It shows that polarization minimization regardless of constraints is similar to convex optimization.

**Weaknesses:**

My main concern is on the contribution of the relaxation and the framework to solve it.
 1. The relaxation seems to be straightforward. It is standard in optimization to relax $\ell_0$-norm into $\ell_1$-norm. And I think it cannot be viewed as a contribution of this work. Other modifications, including replacing the adjacency matrix with Laplacian (then the variable in the loss function and in the constraint become the same), as well as relaxing the constraint from $\le 2k$ to $\le 4k$, are also very slight, from my point of view.
2. What is the contribution of the proposed framework to solve this problem? It seems to be the use of projected gradient descent. But I think the projected gradient descent is also very standard in optimization. So what is the novelty of this method?
3. It is my first time to see polarization minimization. So my confidence is only 2.

**Questions:**

See the weakness part.

**Details Of Ethics Concerns:**

n/a.

---

> ### Author Response · Authors · 2023-11-13
> **Clarifications**
>
> Firstly, we thank the reviewer for taking their valuable time to provide feedback.
>
> Running first-order optimization algorithms on non-convex problems only guarantees us a local minimum. Studying the properties of these local minima and proving an even stronger result that there exists a global minima improves our theoretical understanding of these problems. This, in turn, motivates the design of efficient algorithms for such problems.   There is a wide range of non-convex practical problems which have been studied in this context. These include, but are not limited to, principal components analysis, canonical correlation analysis, orthogonal tensor decomposition, phase retrieval, dictionary learning, matrix sensing, matrix completion, and other nonconvex low-rank problems. A few references are provided below:
>
> Lee, J.D., Simchowitz, M., Jordan, M.I. and Recht, B., 2016, June. Gradient descent only converges to minimizers. In Conference on learning theory (pp. 1246-1257). PMLR.
>
> Ge, R., Huang, F., Jin, C. and Yuan, Y., 2015, June. Escaping from saddle points—online stochastic gradient for tensor decomposition. In Conference on learning theory (pp. 797-842). PMLR.
>
> Our contribution here is similar in this aspect, where we theoretically demonstrate the characteristics of polarization function under the popular  FJ dynamics, which is believed to be non-convex. By showing that polarization (and the general class of $M \rightarrow$ $s^TM^{-k}s$ functions) are invex, a special class of non-convex functions, we ensure there are no saddle points, and every local minimum is a global minimum for this class of functions. This information is valuable in algorithm design as it ensures that first-order methods are not stuck at local minima or saddle points. The objective function for $k=2$, $M \rightarrow$ $s^TM^{-2}s$, is conjectured to have only one local minimum [Citation: Xi Chen et.al., Page 10]. Ours is the first theoretical result that validates this conjecture.
>
> Not only do we theoretically demonstrate the global optimality scenarios for various practical cases of polarization, including stubborn actors and multi-period polarization, but we also deal with scenarios where only partial information about initial opinions is available. Our relaxations provided in equations (5), (6), and (24) aim to identify the edges that provably minimize polarization.
>
> Also, observe that most of the literature is focussed on minimizing convex objectives such as Polarization and Disagreement index or understanding Disagreement among users (references are provided in Section 3: Prior Work).
>
> [Citation: Miklos Z. Rácz et al., page 12] studied this problem in a discrete setting and provided heuristics to minimize polarization.
> Ours is the first continuous optimization study that provides theoretical guarantees in this aspect and outperforms the existing approaches.
>
> Thus, our contributions are multi-fold:
>
> 1) Rather than modelling the problem in a discrete setting, we provide a continuous relaxation backed up with theoretical guarantees for polarization under known initial opinions, including the realistic cases of stubborn actors and multi-period.
> 2) We also study a realistic problem setting that has not been previously studied, wherein the observer has access to only a subset of initial opinions. We show that in these cases, too, PGD can attain the global minimum. Note that this scenario can also be extended to stubborn actors and multi-period.
>
>
> We hope to have addressed the reviewer's questions. We would be happy to answer any further questions.

---

### Official Review · Reviewer_pnnF · 2023-10-27

**Soundness:** 4 excellent
**Presentation:** 3 good
**Contribution:** 3 good
**Rating:** 8
**Confidence:** 3

**Summary:**

The authors propose a new approach for two problems related reducing polarization in a network. In the one variant, opinions are assumed to be observed for all participants in the network, while in another variant, only a subset of opinions are observed. There are assumed to be weights between pairs of users that can be modified by the social network platform, and opinions are assumed to evolve via the Friedkin-Johnsen model. The goal is to minimize the polarization of the network by making changes to the weights of the network, subject to a budget constraint. The authors show that polarization is an invex function, and develop an invex relaxation approach to solve this problem. Computational results are presented on both synthetic and real data.

**Strengths:**

The method provided by the authors is original, and addresses an interesting problem. The computational experiments are reasonable, and demonstrate that the method provides value.The paper is mostly clearly written, other than a couple of points that I mention in the weaknesses.

**Weaknesses:**

It was unclear to me exactly which optimization problem the authors are trying to solve. Is it problem (3) or is it problem (5)? The problem (5) is presented as a relaxation of problem (3), so I am assuming that this work is ultimately intended to solve problem (3). However, as far as I can tell, the procedure proposed by the authors does not guarantee that the resulting solution is feasible for problem (3). The authors should clarify this.

Some of the content presented in the paper seems superfluous, including the material related to polarization under stubbornness and multi-period polarization.

The assumptions that the authors make about the distribution of the unknown opinions seems to be quite strong. The authors could make their work stronger by providing stronger justification for this assumption or by examining how this assumption affects their results. For example, the authors could provide computational experiments where these assumptions are violated.

The authors do not report required computational time of their method.

The computational experiments in the case where some opinions are unknown could be stronger. The only comparison method that the authors provide is one that ignores all known opinions. It would be good to also apply some of the other existing methods, such as the coordinate descent approach where the unknown opinions are mean imputated.

**Questions:**

What, exactly is the optimization problem that you are trying to solve?
If you are trying to solve problem (3), how do you ensure feasibility?

---

> ### Author Response · Authors · 2023-11-13
> **Clarifications**
>
> Firstly, we thank the reviewer for taking their valuable time to provide feedback.
>
> We are solving the problem (5), which is the relaxation of the problem (3). We have now mentioned this explicitly in our paper (page 7, below equation 6 in the rebuttal edition of the paper).
>
> **Regarding the distribution of unknown opinions:** We have provided new proof in Appendix F where the invexity is attained without any assumptions on the distribution of initial opinions (in the rebuttal version of the paper). Thus, the global minimum guarantees are maintained even under the unknown distribution of initial opinions.
>
> We are currently working on the suggested experimentation for the baseline coordinate descent approach where the unknown opinions are mean imputed. We will add a follow-up to this update soon.
>
> We hope to have addressed the reviewer's questions. We would be happy to answer any further questions.

---

> ### Author Response · Authors · 2023-11-16
> **Experiments**
>
> Dear Reviewer,
>
> We have incorporated experiments on coordinate descent (by mean imputation of opinions) and compared the results with Trace and nonconvex relaxations. These experiments are detailed on page 19 of supplementary material under the section labeled "Additional experimentation for partially observable initial opinions."

---

> ### Comment · Reviewer_pnnF · 2023-11-20
> **Clarifications**
>
> Thank you for the clarifications and additional experiments.
>
> I appreciate the expanded theoretical result showing that the expected objective is invex for arbitrary initial distributions of opinions, which I feel strengthens the paper. However, if I am understanding correctly, you would have to know what the initial distribution is in order to apply your method. In practice, this initial distribution is unlikely to be known precisely, and would be estimated somehow. How sensitive is your method to inaccuracies in estimating this initial distribution?

---

> > ### Author Response · Authors · 2023-11-22
> > **Perturbations to Initial Distribution**
> >
> > Dear Reviewer,
> >
> > We appreciate your valuable feedback. In the absence of information on $s_2$, we have assumed its origin from a distribution to formulate a principled approach for minimizing polarization. It is important to note that the specific characteristics of this distribution and exact covariance matrix may vary based on the application.
> >
> > To assess the impact of inaccuracies in estimating the initial distribution, we conducted numerical experiments using a Stochastic Block Model (SBM) comprising 100 nodes and 1210 edges. Our focus lies only in the phase when initial opinions are unknown, with the objective of minimizing $\text{trace}\langle \Sigma, (I+L)^{-2} \rangle$. In these experiments, we employed a perturbed (estimated) covariance matrix, denoted as $\hat{\Sigma}$. The construction of $\hat{\Sigma}$ involved perturbing randomly selected eigenvalues of $\Sigma$. The subsequent paragraph presents the values for $\min \text{trace}\langle \hat{\Sigma}, (I+L)^{-2} \rangle$ and provides a comparison with the true value $\min \text{trace}\langle \Sigma, (I+L)^{-2} \rangle$.
> >
> > Perturbations were introduced to a subset of eigenvalues (40 eigenvalues are perturbed) by randomly adding or deleting values of $\delta$ (the range of the eigenvalues are [0, 3)). The polarization results for a specified budget (k=10) are analyzed, with consistent trends observed across various budgets, ranging from k=1 to 15. The initial polarization value without perturbation is reported as 0.4117.
> >
> >
> > | $\delta$     | Polarization |
> > | ----------- | ----------- |
> > | 0.001     | 0.4121          |
> > | 0.01       | 0.4112          |
> > | 0.1         | 0.4209          |
> > | 0.25       | 0.3822         |
> > | 0.5         | 0.3243          |
> >
> > We think the potential theoretical implications of minimizing polarization under imprecise specifications of the covariance of unknown initial opinions is an interesting area to explore. This requires a careful analysis and we will duly consider it as our future research direction.

---

### Official Review · Reviewer_XHJ6 · 2023-11-03

**Soundness:** 2 fair
**Presentation:** 2 fair
**Contribution:** 1 poor
**Rating:** 1
**Confidence:** 5

**Summary:**

The paper studies the problem of minimizing polarization in Friedkin-Johnson (FJ) model, where polarization simply measures how close the given network is to consensus. In particular, given an adjacency matrix on an undirected graph, the problem at hand is to find a new adjacency matrix which only differs from the original by a given budget and minimizes the polarization. It is expected that this problem is difficult in nature (due to the sparse/zero norm constraints), which is what is stated. The authors then provide a nonconvex relaxation and show that this relaxation falls into the category of an invex function minimization, and naturally use this to provide a trackable formulation.

**Strengths:**

The paper is not suitable for this venue.

**Weaknesses:**

Regardless of the merits of the contributions, the paper is not suitable for ICLR.

The problem is also not well motivated, and does not appear to be addressing a fundamental issue or question; the problem seems to be defined in a way that its relaxation fits to an invex function minimization problem. The related literature is not well surveyed; there is a wide range of optimization problems on graph Laplacian learning that could be relevant here, and the literature on Friedkin-Johnson (FJ) model is far from complete.

**Questions:**

N/A

---

> ### Author Response · Authors · 2023-11-13
> **Rebuttal**
>
> Dear reviewer,
>
> **Relevance to ICLR:** The URL for the ICLR conference has a “general machine learning” track, which is where we submitted our paper. Also, the conference has non-convex optimization listed in the subject areas. The webpage also says, “A non-exhaustive list of relevant topics.” So, we believe the conference is open to other ML topics, such as ours, where we characterize the objective function followed by a non-convex relaxation pertinent to the problem under study.
>
> We strongly believe that we addressed all the relevant state-of-the-art literature pertinent to the problem in the prior work section. We also provided an additional literature review in Appendix A of the paper that was submitted for review.
>
> **Motivation:** The topic of polarization has been motivated in the introduction section of the paper. The formal Instance of the polarization minimization problem is provided on Page 2 of our paper.  Also, we would like to point out that we are not the first to study the polarization minimization problem. This problem has been initially proposed in the [Citation: Xi Chen et al. on Page 10, Cameron Musco et al. on Page 11 of references].
>
> **Laplacian Learning:** Note that the current literature on Laplacian Learning, such as the ones listed below, does not have the negative exponent of “-2” in their objective, unlike what we have for polarization minimization ($s^T(I+L)^{-2}s$). So, the current literature on Laplacian learning is ***not*** directly applicable.
>
> Changhao Shi and Gal Mishne, 2023. Graph Laplacian Learning with Exponential Family Noise. Fourteenth International Conference on Sampling Theory and Applications.
>
> Egilmez, H.E., Pavez, E. and Ortega, A., 2017. Graph learning from data under Laplacian and structural constraints. IEEE Journal of Selected Topics in Signal Processing, 11(6), pp.825-841.
>
> Kumar, S., Ying, J., de Miranda Cardoso, J.V. and Palomar, D., 2019. Structured graph learning via Laplacian spectral constraints. Advances in neural information processing systems, 32.

---

### Author Response · Authors · 2023-11-16
**Additional Inclusions in the paper**

In the revised supplementary material, we have included additional evidence in Appendix F demonstrating the achievement of invexity without relying on assumptions about the distribution of initial opinions (in the context of partially observable initial opinions). Thus, the guarantees of attaining global minimum are maintained even under the unknown distribution of initial opinions. We have also addressed the comments made by the reviewer pnnF in the main paper.

Additionally, we have incorporated experiments on coordinate descent (by mean imputation of opinions) and compared the results with Trace and nonconvex relaxations. These experiments are detailed on page 19 of supplementary material under section E, "Additional experimentation for partially observable initial opinions."

---

### Meta-Review · Area_Chair_uszu · 2023-12-06

**Metareview:**

While there was no consensus among the reviewers about this paper, I agree with the more negative reviewers that (i) the problem is poorly motivated (ii) given the lack of an explicit complexity result over an arbitrary number of time steps, the result attained are not very strong (iii) the paper is not a good fit for ICLR.

**Justification For Why Not Higher Score:**

See metareview.

**Justification For Why Not Lower Score:**

N/A

---

### Decision · Program_Chairs · 2024-01-16

Reject